# Poly(GR) and poly(GA) in cerebrospinal fluid as potential biomarkers for *C9ORF72*-ALS/FTD

Gopinath Krishnan[1], Denitza Raitcheva[2], Daniel Bartlett[2], Mercedes Prudencio[3], Diane M. McKenna-Yasek[1], Catherine Douthwright[1], Björn E. Oskarsson[4], Shafeeq Ladha[5], Oliver D. King[1], Sami J. Barmada [6], Timothy M. Miller[7], Robert Bowser [5], Jonathan K. Watts [8], Leonard Petrucelli[3], Robert H. Brown [1], Mark W. Kankel[9,10 ✉] & Fen-Biao Gao [1 ✉]

GGGGCC repeat expansion in *C9ORF72*, which can be translated in both sense and antisense directions into five dipeptide repeat (DPR) proteins, including poly(GP), poly(GR), and poly(GA), is the most common genetic cause of amyotrophic lateral sclerosis (ALS) and frontotemporal dementia (FTD). Here we developed sensitive assays that can detect poly(GA) and poly(GR) in the cerebrospinal fluid (CSF) of patients with *C9ORF72* mutations. CSF poly(GA) and poly(GR) levels did not correlate with age at disease onset, disease duration, or rate of decline of ALS Functional Rating Scale, and the average levels of these DPR proteins were similar in symptomatic and pre-symptomatic patients with *C9ORF72* mutations. However, in a patient with *C9ORF72*-ALS who was treated with antisense oligo-nucleotide (ASO) targeting the aberrant *C9ORF72* transcript, CSF poly(GA) and poly(GR) levels decreased approximately 50% within 6 weeks, indicating they may serve as sensitive fluid-based biomarkers in studies directed against the production of GGGGCC repeat RNAs or DPR proteins.

[1] Department of Neurology, University of Massachusetts Chan Medical School, Worcester, MA 01605, USA. [2] Biomarkers, Clinical Sciences Biogen, Cambridge, MA 02142, USA. [3] Department of Neuroscience, Mayo Clinic, Jacksonville, FL 32224, USA. [4] Department of Neurology, Mayo Clinic, Jacksonville, FL 32224, USA. [5] Departments of Neurology and Translational Neuroscience, St. Joseph's Hospital and Medical Center and Barrow Neurological Institute, 350W Thomas Road, Phoenix, AZ 85013, USA. [6] Department of Neurology, University of Michigan, 4005 BSRB, 109 Zina Pitcher Place, Ann Arbor, MI 48109-2200, USA. [7] Department of Neurology, Washington University, Saint Louis, MI 63110, USA. [8] RNA Therapeutics Institute and Department of Biochemistry and Molecular Pharmacology, UMass Chan Medical School, Worcester, MA 01605, USA. [9] Neuromuscular & Movement Disorders, Biogen, Cambridge, MA 02142, USA. [10] Present address: Apple Tree Partners (ATP) Research Labs, Branford, CT 06405, USA. ✉email: mkankel@atpresearchlabs.com; fen-biao.gao@umassmed.edu

Frontotemporal dementia (FTD) and the motor neuron disease amyotrophic lateral sclerosis (ALS) are regarded as spectrum disorders whose clinical, pathological, and genetic characteristics overlap substantially[1–3]. For instance, the most common known genetic cause of both diseases is a GGGGCC ($G_4C_2$) repeat expansion in the first intron of *C9ORF72*[4,5]. RNAs transcribed from these repeats in both sense and antisense directions can be translated into five dipeptide repeat (DPR) proteins—poly(GA), poly(GR), poly(GP), poly(PR), and poly(PA)[6–8]. Among them, poly(GR) is likely to be a key neurotoxic species, as its expression strongly correlates with neurodegeneration in the brains of patients with *C9ORF72* mutations[9–11]. Moreover, poly(GR) is toxic in many cellular and animal models[12,13]. For example, we established an inducible mouse model in which $(GR)_{80}$ is expressed in the brain at ~5–15% of the level in the brains of patients with *C9ORF72* mutations and demonstrated that low-level poly(GR) expression alone can elicit several FTD/ALS-like behavioural and cellular phenotypes[14]. Poly(GA), the most abundantly expressed DPR protein in patients[15,16], is likely to be another pathogenic DPR protein, as mouse models of poly(GA) toxicity exhibited behavioural deficits, neuronal cell loss, and TDP-43 pathology[17,18]. Similarly, poly(GA) expression in primary neurons or HEK293T cells induces the aggregate formation and cellular toxicity[19,20] and disease-related TDP-43 cleavage[21].

Although not overly toxic in cellular and animal models, poly(GP) is widespread in patient brains with *C9ORF72* mutations and detectable in their cerebrospinal fluid (CSF), but the levels do not correlate with disease onset or clinical scores and stay relatively constant over time[22–24]. To date, it is unknown whether other DPR proteins such as poly(GA) and poly(GR) can also be detected in CSF from patients with *C9ORF72* mutations and if so, whether the levels correlate with any clinical features of patients. Although poly(GR) and poly(GA) are neurotoxic in cellular and animal models, it is not known whether these DPR proteins can serve as diagnostic or target engagement biomarkers in clinical trials.

One promising therapeutic approach for *C9ORF72*-ALS/FTD is to use antisense oligonucleotides (ASOs) to reduce the expression of $G_4C_2$ repeat-containing RNAs. ASOs have been used successfully to alter splicing patterns in spinal muscular atrophy (SMA) and hold great promise for treating other neurodegenerative diseases[25]. Indeed, an ASO targeting the first intron of *C9ORF72* has been given to a single patient with ALS[26]. Another potential therapeutic approach is antibodies against specific DPR proteins. In a study of *C9ORF72* human BAC transgenic mice, targeting poly(GA) proteins with an antibody reduced poly(GA) aggregates and increased survival, improved behaviour, and rescued neurodegeneration[27]. Additionally, active poly(GA) vaccination prevents microglia activation and motor deficits in mice overexpressing poly(GA)[28]. For these and other therapies targeting $G_4C_2$ repeat RNAs or DPR proteins, it is essential to develop assays to detect poly(GA) and poly(GR) in human fluids. Here we describe our efforts to establish poly(GA) and poly(GR) in CSF as sensitive biomarkers of *C9ORF72*-ALS/FTD.

## Results

**Poly(GR) is a CSF biomarker specific for patients with *C9ORF72*-related ALS/FTD.** We previously established a Meso Scale Discovery (MSD)-based assay to measure poly(GR) expression in transgenic mice and flies as well as in human neurons differentiated from induced pluripotent stem cell (iPSC) lines of patients with *C9ORF72* mutations[14,29,30]. MSD immunoassay is a

highly sensitive method that uses electrochemiluminescence (ECL) to detect protein levels, in contrast to the less sensitive colorimetric reaction used in traditional enzyme-linked immunosorbent assays (ELISAs)[31]. To establish that our MSD immunoassay can detect poly(GR) in CSF from patients with *C9ORF72* mutations, we first performed spike-and-recovery experiments in which synthetic $(GR)_8$ peptide was added to CSF samples from healthy people at concentrations of 1.56–25 pg/ml; the recovery rate was 75–92% across five different concentrations (Fig. 1a). The standard curve of $(GR)_8$ peptide was 1.56–100 pg/ml; 3 pg/ml was the lower limit of quantification (LLOQ) (Fig. 1b). The coefficient of variance (CV) was 0.67–12.81% for intraplate replicates, 1.54–15.01% for interplate replicates, and 1.89–20.69% for day-to-day replicates (Fig. 1c). Assays with a CV < 20% are considered reliable by the US Food and Drug Administration[32]. To demonstrate the clinical utility of the assay as recommended[33], we used CSF samples from three patients with *C9ORF72* mutations. The intra-assay CV was 1.35–0.50% and the inter-assay CV was 7.67–16.24% (Supplementary Fig. 1a). CSF samples from 25 patients with *C9ORF72* mutations measured in duplicate well in a single experiment have an intra-assay CV between 0.15 and 20.35%. (Supplementary Fig 1b). Thus, the assay meets acceptable quality control statistical standards. Next, we did a dilution linearity test using two patient samples spiked with 200 ng/ml of $(GR)_8$ peptide. The dose-response was linear (with R square values of 0.987 and 0.997) within the range of the standard curve (Fig. 1d). Finally, we assessed GR immunoassay selectivity by testing up to 100 ng/ml $(GP)_8$ and $(GA)_8$ peptides spiked in control CSF samples. We found no significant signal, suggesting that the assay specifically detects $(GR)_8$ (Fig. 1e).

To further demonstrate that we can reliably measure poly(GR) in a blinded manner, we used our MSD assay to examine CSF samples from two independent cohorts, one consisting of four healthy and four samples from patients with *C9ORF72* mutations and the other consisting of three samples from healthy participants and five samples from patients with *C9ORF72* mutations. In healthy CSF samples, poly(GR) was not detected, whereas a clear signal above the background was observed for poly(GR) in the *C9ORF72* CSF samples (Fig. 1f, g). Similar results were obtained with CSF samples from a third, larger cohort of CSF samples from 19 healthy participants, 19 patients with ALS but without *C9ORF72* mutations, and 10 patients with *C9ORF72*-ALS from the NEALS Consortium (Fig. 1h). None of the CSF samples from the NEALS Consortium was excluded due to a low poly(GR) signal in our assay. Thus, our results show that poly(GR) is detectable in CSF and is a potentially valid fluid-based biomarker for *C9ORF72*-ALS/FTD.

**Poly(GR) and poly(GA) levels in CSF do not correlate with clinical features of patients with *C9ORF72* mutations.** Since poly(GR) expression in patient brains with *C9ORF72* mutations correlates with neurodegeneration and clinical phenotypes[9–11], we examined whether poly(GR) levels in CSF samples of patients with *C9ORF72*-related ALS and ALS/FTD correlate with clinical features. For this analysis, we used a combined set of CSF samples from the *C9ORF72* Natural History and Biomarkers Study (PMID 31578300) and the NIH Intramural study NCT01925196 (Biogen cohort). The Revised Amyotrophic Lateral Sclerosis Functional Rating Scale (ALSFRS-R) scores and age of onset and duration of disease were available for all patients with *C9ORF72* mutations for whose CSF samples were used here; the samples were further classified according to whether the patients were symptomatic or pre-symptomatic. We obtained 41 CSF samples from symptomatic patients with *C9ORF72* mutations and 14 CSF samples from pre-symptomatic *C9ORF72* carriers. In samples from

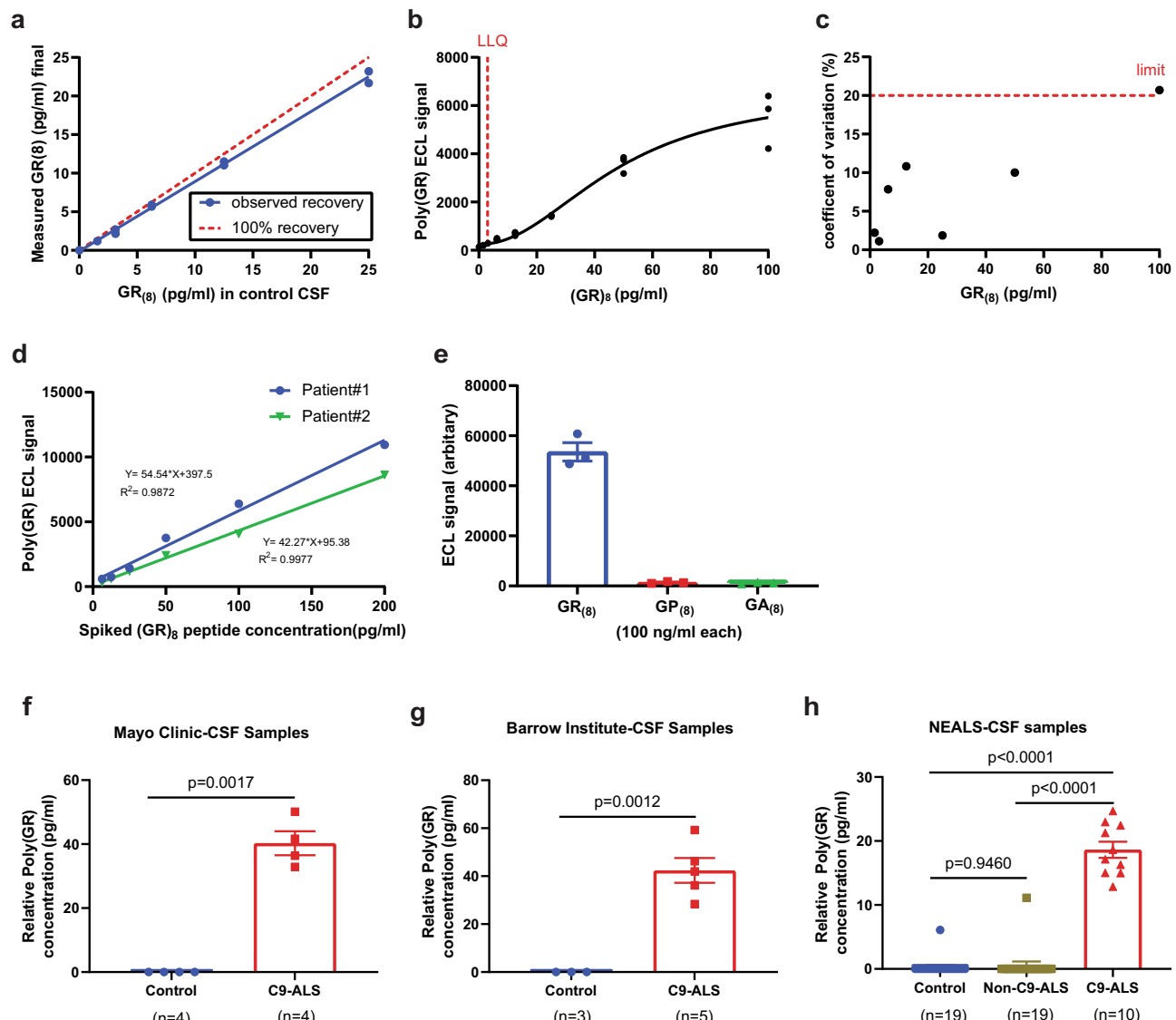

**Fig. 1 Blinded measurement of poly(GR) levels in CSF in multiple cohorts of patients with *C9ORF72* mutations using the MSD-based poly(GR) immunoassay. a** In spike-and-recovery experiments, recombinant $(GR)_8$ peptide was added to CSF from healthy individuals and subsequently serially diluted into a separate tube containing CSF from healthy individuals to create a 5-point series. The average recovery rate in two different samples was 81–88%. **b** The assay can detect purified $(GR)_8$ peptide at concentrations as low as 3 pg/ml. Replicates of 8-point series standard curve run on different dates are shown. A four-parameter logistic curve was used to fit the dose-response with GraphPad Prism 9.1. **c** Across the dynamic range of the assay, the coefficient of variation is less than 16% for all points except one concentration (100 pg/ml), below the 20% cutoff recommended by the FDA in standard variability for liquid-binding assays. **d** Dilution linearity was evaluated by measuring duplicates of 5 dilutions of 2 patient CSF samples spiked with 200 ng/ ml $(GR)_8$ peptide. **e** Poly(GR) immunoassay specifically detects $(GR)_8$ peptide in control CSF and did not cross react with even up to 100 ng of $(GP)_8$ peptide in control CSF, $n = 3$ independent experiments, values are mean ± s.e.m. **f** CSF samples from 4 healthy and 4 patients with *C9ORF72* mutations from the Mayo Clinic, Florida, values are mean ± s.e.m ($p = 0.0017$, Unpaired *t*-test with Welch's correction). **g** CSF samples from 3 healthy people from the NEALS cohort and 5 CSF samples from two patients with *C9ORF72* mutations from the Barrow Institute, Arizona, values are mean ± s.e.m ($p = 0.0012$, Unpaired *t*-test with Welch's correction). **h** 19 healthy people, 19 patients with ALS but without *C9ORF72* mutations (3 patients with duplicate samples, 16 patients with single samples), and 10 patients with *C9ORF72* mutations (8 patients with duplicate samples, 2 patients with single samples) from the NEALS Consortium. Values from duplicate samples are presented as mean ± s.e.m ($p = 0.9460$ control vs non-C9-ALS, $p < 0.0001$ non-C9-ALS vs C9-ALS, $p < 0.0001$ control vs C9-ALS Ordinary one-way ANOVA-Tukey's multiple comparisons test). Source data are provided as a Source Data file.

25 symptomatic patients that had poly(GR) levels within the linear range, poly(GR) levels in CSF did not correlate significantly with ALSFRS-R scores (Supplementary Fig. 2a) or the rate of ALSFRS-R score decline (Fig. 2a). Nor did poly(GR) levels in CSF correlate with the age of onset (Fig. 2b) or duration of disease (Fig. 2c). Moreover, CSF samples from 10 pre-symptomatic patients with *C9ORF72* mutations had poly(GR) levels within the linear range and did not differ significantly from the levels in samples from symptomatic patients (Fig. 2d). Additional CSF samples especially from pre-symptomatic patients are needed to further confirm this result. Similar analyses of a subset of patient samples from this combined cohort with detectable poly(GA) levels within the linear range of our $(GA)_{60}$ standard curve also showed no correlation between CSF poly(GA) and ALSFRS-R scores (Supplementary Fig. 2b), the rate of ALSFRS-R score decline (Fig. 3a), age of onset (Fig. 3b), or disease duration

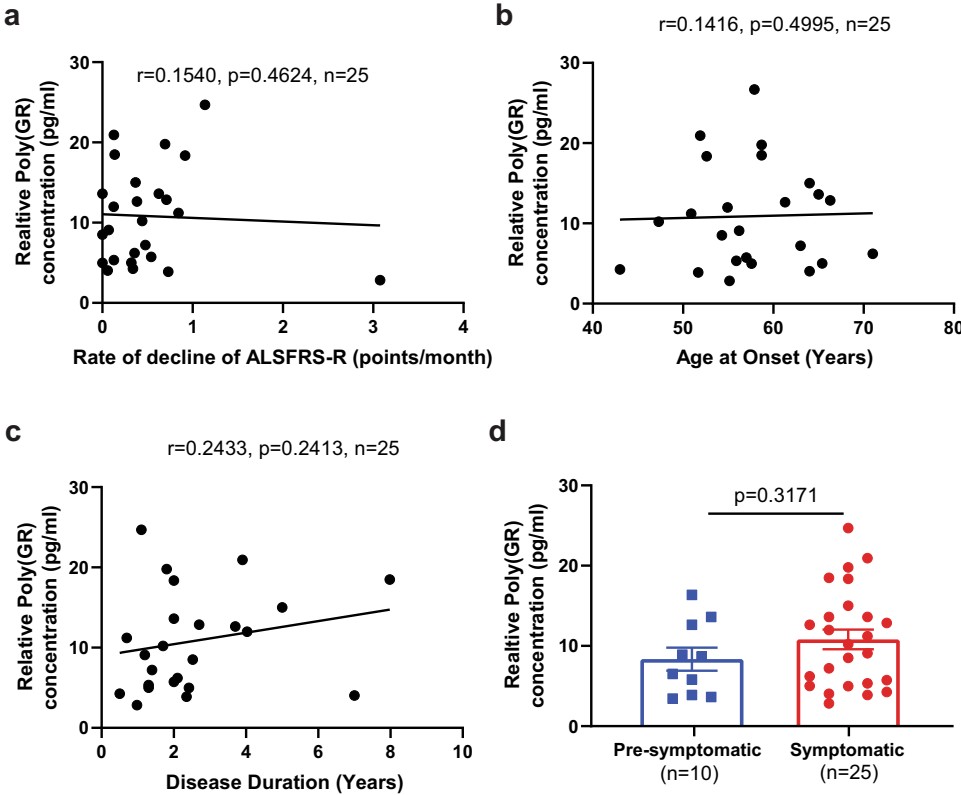

**Fig. 2 Poly(GR) levels in CSF do not correlate with clinical features of patients with *C9ORF72* mutations. a–c** Correlations between poly(GR) levels and rate of ALSFRS-R score decline (**a**), age at disease onset (**b**), and disease duration (**c**), determined by two-tailed, Spearman's rank-correlation analysis. **d** Poly(GR) levels in CSF of symptomatic and asymptomatic patients with *C9ORF72* mutations are similar ($p = 0.114$ by two-tailed unpaired *t*-test with Mann–Whitney test). Poly(GR) concentrations are presented as values equivalent to $(GR)_8$ peptide and not as an absolute poly(GR) concentration since the size of poly(GR) in CSF is unknown. 25 out of 41 symptomatic CSF samples and 10 out of 14 pre-symptomatic CSF samples had poly(GR) levels within the linear range of our standard curves and are presented here. ALS Functional Rating Scale revised (ALSFRS-R). Source data are provided as a Source Data file.

(Fig. 3c). The samples assessed in Figs. 2, 3 are overlapping but not identical and include 17 samples in which both GA and GR were detected. Finally, CSF poly(GA) levels in CSF did not differ between symptomatic and pre-symptomatic patients (Fig. 3d).

**Poly(GR) and poly(GA) in CSF may serve as biomarkers for ASO treatment in patients with *C9ORF72* mutations.** To determine whether poly(GR) or poly(GA) may serve as a potential target engagement biomarker, we must first understand whether their CSF levels change or remain constant over time. We found that CSF poly(GR) levels in longitudinal samples from two symptomatic and two asymptomatic patients with *C9ORF72* mutations were stable over 18–54 months (Fig. 4a). This finding was confirmed for both poly(GR) and poly(GA) in a second, larger cohort (Fig. 4b, c).

Next, we assessed whether changes in poly(GR) and poly(GA) CSF levels could be detected in a patient with *C9ORF72*-ALS who was treated with an ASO targeting the sense strand of the first intron of *C9ORF72* in a recently published clinical trial[26]. Longitudinal CSF samples were obtained at multiple time points after ASO treatment[26]. Six weeks after the first ASO treatment, we found that poly(GP) CSF levels decreased only by about 10%, whereas poly(GR) and poly(GA) CSF levels decreased by 40–50%, suggesting a substantially faster initial kinetics of decline at least in this patient with *C9ORF72*-ALS (Fig. 4d). Moreover, within 36 weeks after multiple ASO treatments, the CSF levels of all three DPR proteins decreased to comparable levels at or below 20% of original levels (Fig. 4d). Thus, if confirmed in additional

patients with *C9ORF72*-ALS/FTD, poly(GR) and poly(GA) may be better target engagement biomarkers than poly(GP) because of their more rapid kinetics of decline.

## Discussion

As $G_4C_2$ repeat expansion in *C9ORF72* is the most common genetic cause of both ALS and FTD, significant effort is ongoing to develop therapies to reduce repeat-containing RNAs and their translation products such as poly(GA). Given the emphasis of these approaches for clinical trials, the ability to measure target engagement is necessary to properly evaluate the results of these trials. Thus, we determined whether poly(GA) and poly(GR) can be detected in patient CSF by MSD-based assays and whether CSF poly(GA) and poly(GR) levels correlate with clinical measures and could therefore serve as biomarkers of therapeutic efficacy. Finally, we assessed CSF poly(GA) and poly(GR) levels for their ability to be used as potential biomarkers. We found that neither poly(GA) nor poly(GR) in CSF correlated with clinical measures, suggesting that their levels in CSF may not be useful for assessing disease severity or therapeutic efficacy. However, the CSF levels of both DPR proteins decreased rapidly in a single patient with *C9ORF72*-ALS treated with an ASO targeting the intron-containing $G_4C_2$ repeats. If confirmed in additional patients, poly(GR) and poly(GA) may serve as biomarkers of target engagement.

The lack of correlation between the level of poly(GA)/poly(GR) and clinical features of patients with *C9ORF72* mutations mirrors what was reported for poly(GP), another DPR protein detectable

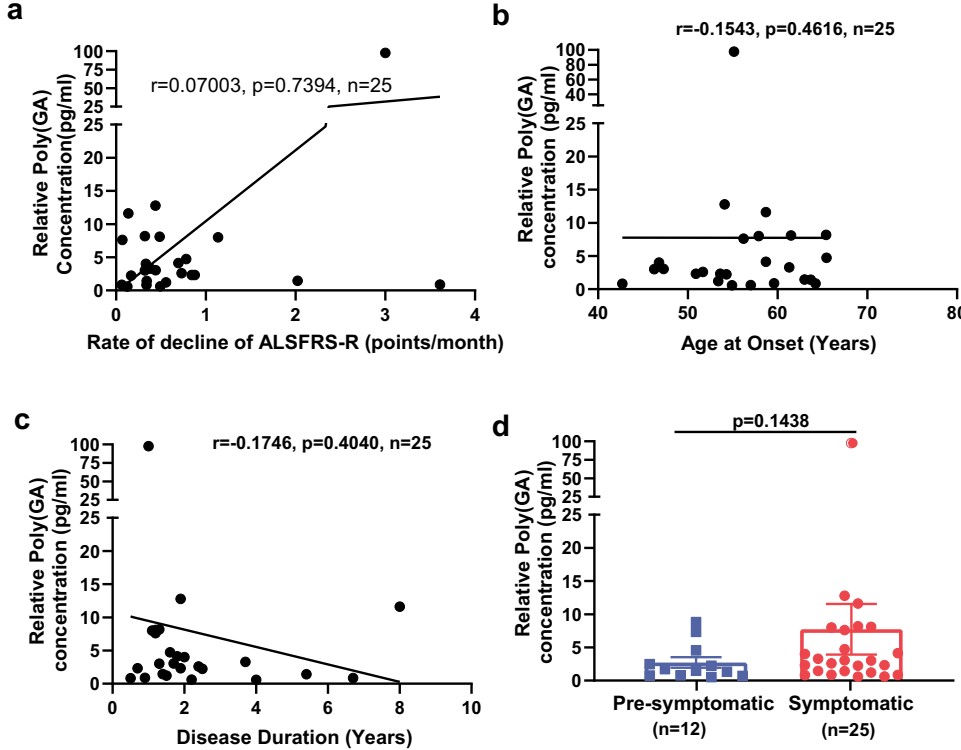

**Fig. 3 Poly(GA) levels in CSF do not correlate with clinical features of patients with *C9ORF72* mutations. a–c** Correlation between poly(GA) levels and the rate of ALSFRS-R score decline (**a**), age at disease onset (**b**), and disease duration (**c**), determined by two-tailed Spearman's rank-correlation analysis. **d** Poly(GA) levels in CSF from symptomatic and asymptomatic patients with *C9ORF72* mutations are similar by two-tailed unpaired *t*-test with Mann–Whitney test. Poly(GA) concentrations are presented as values equivalent to $(GA)_{60}$ and not as an absolute poly(GA) concentration since the size of poly(GA) in CSF is unknown. The samples assessed in Figs. 2, 3 are overlapping but not identical and include 17 samples for which both poly(GA) and poly(GR) were detected. Source data are provided as a Source Data file.

in CSF whose levels do not correlate with disease onset or clinical scores and stay relatively constant during disease progression[22–24,34–36]. Together, these results do not rule out the possibility that DPR proteins are important in the pathogenesis of *C9ORF72*-related diseases, since the dynamic relationship between DPR protein levels in specific vulnerable neurons and in CSF is not well established. Moreover, additional studies are needed to investigate post-translational modifications of DPR proteins (e.g. methylation) and their effects on their neurotoxicity and detections in CSF samples.

Poly(GP) is currently used as a target engagement biomarker for *C9ORF72* therapies targeting $G_4C_2$-repeat-containing transcripts because it is expressed in patient brains with *C9ORF72* mutations, is present in patient CSF, and correlates with levels of repeat-containing RNA in the cerebellum of *C9ORF72* carriers[22–24,34–36]. Our findings show that poly(GA) and poly(GR) could also potentially serve as target engagement biomarkers, as their levels remain relatively constant over time and appear to respond more quickly to ASO therapy than poly(GP). We attribute the slow decrease in CSF poly(GP) to its expression from both sense and antisense repeat RNAs and widespread accumulation due to its demonstrated lack of neuronal toxicity in the brain.

Our study has some limitations that are worth discussing. First, several CSF samples from the Biogen cohort exhibited poly(GR) or poly(GA) signals outside the linear range of our standard curves and therefore could not be included in our analyses described here. In contrast, we were able to measure poly(GR) levels for all of the NEALS cohort CSF samples. Given this observation, we speculate that there are likely to be unknown factors that contribute to CSF DPR protein levels and may

therefore account for these differences. For instance, it is possible that the collection, processing and storage of the CSF samples is a key aspect to sample integrity and that different clinical sites have slightly modified protocols that may affect sample quality (e.g. the length of collection time, the person collecting samples, collection vessels, time, and site of puncture, etc.). Consistent with this notion, others have shown that variations in sample handling can affect the measurable levels of prion protein in the CSF[37]. Nevertheless, given that detectable levels of both poly(GA) and poly(GR) were observed in a majority of cases, in addition to further improvement of assay sensitivity, one way to overcome this limitation in a clinical trial setting is to first ensure the quality of CSF samples and then pre-screen trial participants based on detectable CSF DPR protein load prior to enrollment. With such patient selection criteria used in trial designs, appropriate assessment of candidate therapeutics will become more robust. Second, we only managed to get access to small numbers of CSF samples for some analyses; for instance, the number of pre-symptomatic CSF samples is relatively low, and longitudinal CSF samples were available only from a single patient with *C9ORF72*-ALS after compassionate use of experimental ASO treatment[26]. The faster decrease of CSF poly(GA) and poly(GR) levels than of the poly(GP) level needs to be confirmed in additional patients with *C9ORF72* mutations after ASO treatment. However, our findings that poly(GA) and poly(GR) can be detected in good quality CSF samples and that their levels decrease in response to ASO treatment suggest the potential for these DPR proteins to serve as sensitive target engagement biomarkers in the development of *C9ORF72*-directed therapies. For instance, these biomarkers could be used to assess correlations between improved clinical features and loss of DPR proteins, and in prolonged

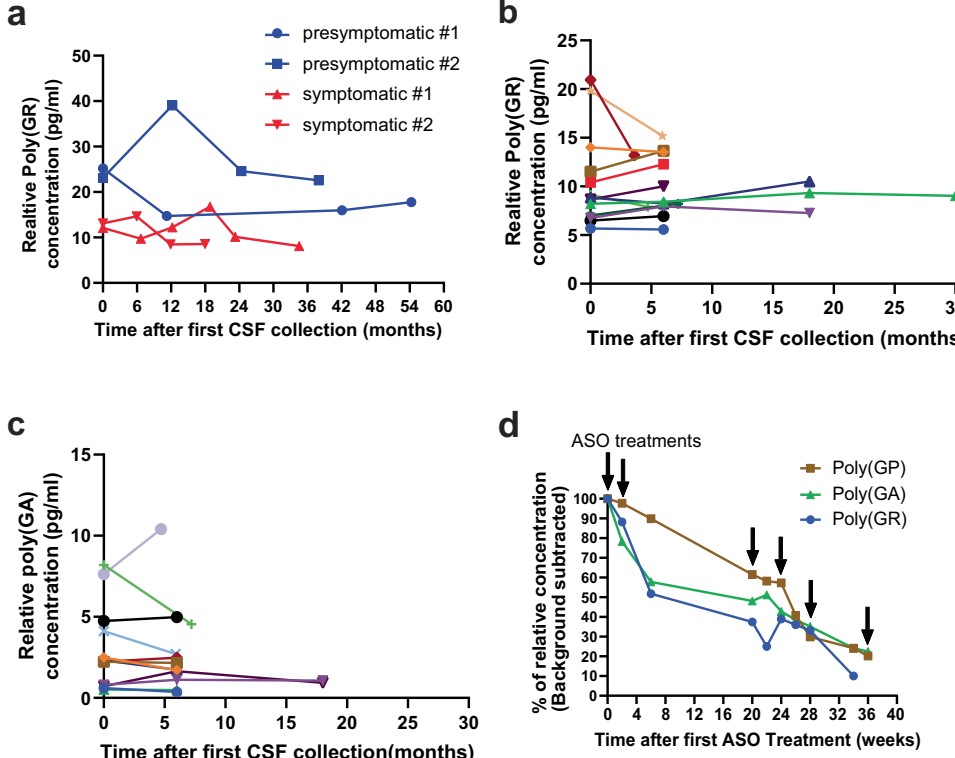

**Fig. 4 ASO treatment in a patient with *C9ORF72*-ALS decreases CSF DPR levels. a** Poly(GR) levels in CSF were generally stable over time in two symptomatic and two asymptomatic patients with *C9ORF72* mutations. **b** Longitudinal trend of poly(GR) levels in CSF from a larger cohort of patients with *C9ORF72* mutations. **c** Longitudinal trend of poly(GA) levels in CSF from patients with *C9ORF72* mutations. **d** After ASO treatment, CSF poly(GP), poly(GA), and poly(GR) levels in a patient with *C9ORF72*-ALS decreased over time. Note: colors in panels **b**, **c** do not indicate the same patients. The fifth time point value for symptomatic patient 2 in panel **a** and the last time point for poly(GR) value in panel **b** were above background but below the lower limit of detection and were not included. Source data are provided as a Source Data file.

clinical trials, they could be used to show that when a lack of efficacy or a negative outcome occurs, it is not due to the lack of target engagement.

## Methods

**CSF samples**. Research complies with all relevant ethical regulations; Patients gave written informed consent for sharing de-identified biospecimens under protocols reviewed by the NIH Institutional review board (NCT01925196), Massachusetts General Hospital (2015P002295, 2015P002034, 2011P000511, and 2007P002586/1), UMass Chan Medical School (IRB docket #14341 and #H00004956), Mayo Clinic Jacksonville (13-004314), St. Joseph's Hospital, and Medical Center (IRB#11BN125). ASO treatment of a patient with *C9ORF72*-ALS was performed at UMass Chan Medical School (IRB docket #14341 and #H00004956). Coded CSF samples from healthy people and patients with *C9ORF72* mutations were obtained from multiple institutions including some previously published samples[22,24,38]. Requests for these samples should be addressed to these sources. The clinical features of the patients are summarized in Supplementary Tables 1 and 2. The rate of decline of ALSFRS-R scores was calculated as 48 minus ALSFRS-R score at sampling divided by disease duration in months.

**Poly(GR) immunoassay of CSF samples**. Poly(GR) levels in CSF samples from healthy people and patients with *C9ORF72* mutations were measured in a blinded manner, with custom-made, affinity-purified rabbit polyclonal GR antibodies raised against (GR)₈ peptide and a Meso Scale Discovery platform that uses electrochemiluminescence detection technology as described[14] with minor modifications. In brief, biotinylated poly(GR) antibodies were diluted in phosphate-buffered saline (PBS) at a concentration of 0.5 µg/ml and coated on 96-well single-spot streptavidin plates (MSD Gold) and incubated overnight at 4 °C. The next day, antibodies were removed, washed three times with Tris-buffered saline (TBS)-Tween20 (0.05%) wash buffer, and blocked with 1% BSA-TBS-Tween20 (0.05%) blocking solution or MSD Blocker A solution for 1 h. CSF samples (150–200 µl) were concentrated with Amicon Ultra 0.5 ml 3 kDa filters by centrifugation at 14,000 × *g* for 15 min at room temperature. After 3 kDa filter centrifugation, very small dipeptides may flow through the column but we do not expect that pathological poly(GR) proteins are smaller than 3 kDa. Concentrated CSF samples

(45 µl) were loaded in duplicate wells and incubated for 1.5 h on a shaking platform. After three washes, plates were loaded with MSD-Gold-Sulfo-tagged poly(GR) detection antibody (0.5 µg/ml), incubated for 1 h at room temperature on a shaking platform, and washed three times. MSD-Read buffer (1×) was added, and the plates were immediately read with an MSD-QuickPlex SQ 120 Reader. After background correction, sample electrochemiluminescence (ECL) signals were interpolated against a standard curve prepared with serial dilutions of recombinant (GR)₈ peptide and presented as relative concentration. The lower limit of detection was calculated using the mean of the background plus 2.5 times the standard deviation. The sizes of DPR proteins in CSF are unknown; we measured the amount of total poly(GR) and poly(GA) equivalent to the epitopes detected in (GR)₈ and (GA)₆₀ based on standard curves.

**Poly(GA) immunoassay analysis for CSF samples**. Poly(GA) levels in *C9ORF72* derived patient CSF were measured with a sandwich immunoassay assay that utilizes the same antibody combinations used to detect poly(GA) levels in several prior publications[27,30,39,40] and the MSD ultrasensitive S-PLEX platform. The MSD S-PLEX assay utilizes a modified version of the standard MSD assay protocol, which includes an alternate detection antibody label, TURBO-BOOST and an enhancement step featuring a TURBO-TAG label to boost ECL signal and improve assay sensitivity. Briefly, undiluted CSF was tested for poly(GA) by using human/murine chimeric forms of capture and detection antibodies as described[27]; however, the detection antibody was further modified with a TURBO-BOOST conjugation. Each sample was tested in duplicate wells, sample volume permitting, using 25 µl per well in MSD S-PLEX 96-well SECTOR plates. The 8-point calibration curve was prepared by serially diluting (GA)₆₀ protein expressed in HEK 293 cells, and four QC samples were included on each assay plate. Electrochemical stimulation of the assay plate with the MESO® SECTOR S 600 instrument generates ECL signal values that correspond to the intensity of emitted light. The ECL signal level is proportional to the amount of poly(GA) in the sample, and the concentrations in CSF samples were interpolated from the calibration curve.

**Statistics**. Nonparametric Spearman's rank-correlation tests were used for all correlation analyses. Nonparametric unpaired *t*-tests and Mann–Whitney tests were used to compare samples from symptomatic vs. asymptomatic patients.

Statistical analyses of poly(GR) and poly(GA) were done with Prism v9.1 and Prism v7.02., respectively.

**Reporting summary**. Further information on research design is available in the Nature Research Reporting Summary linked to this article.

## Data availability

All the data presented in this study are included in the Source Data File. Requests for patient CSF samples should be addressed to the respective institutions.

## Code availability

Not applicable.

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

## Acknowledgements

We thank many patients and their families for the CSF samples used in this study. The NIH Intramural study NCT01925196 led by Dr. Mary Kay Floeter and the NEALS Biorepository led by Dr. James D. Berry provided some CSF samples from patients with ALS and healthy people. The detection and capture antibodies used to evaluate CSF poly(GA) levels were discovered at Neurimmune. This work was supported by grants from the Association for Frontotemporal Degeneration and the Target ALS Foundation (F.-B.G., M.K., S.J.B., and R.B.), the NIH (R37NS057553 and R01NS101986 to F.-B.G., R01NS111990 to R.H.B. and J.K.W.; R01NS078398 to T.M.M.; R35NS097273, P01NS084974, and P01NS099114 to L.P.), and the Angel Fund for ALS Research (R.H.B.) ALS Association (T.M.M.), Biogen (T.M.M.).

## Author contributions

F.B.G. and M.W.K. coordinated the project. G.K. performed all the poly(GR) measurements and statistical analyses and made figures for poly(GP) and poly(GR). D.R. and D.B. were responsible for poly(GA) measurement and statistical analyses and prepared drafts for the poly(GA) figures. M.P., D.M.M.Y., C.D., B.E.O., S.L., T.M.M., R.B., J.K.W., L.P., and R.H.B. provided CSF samples, O.K. helped with statistical analysis, S.J.B. was involved in aspects of data analysis and interpretation. G.K., M.W.K., and F.B.G.

wrote the paper with inputs from other coauthors. All authors read and approved the final manuscript.

## Competing interests

D.R., D.B., and M.W.K. are employees and shareholders of Biogen. S.L. received research support from Biogen (Cambridge, MA), Sanofi (Paris, France), Amylyx Pharmaceuticals (Cambridge, MA), Mitsubishi Tanabe Pharma (Osaka, Japan), and consulting from Biogen (Cambridge, MA). B.O. received research support from Biogen (Cambridge, MA), Mitsubishi Tanabe Pharma America (Jersey City, NJ), MediciNova (La Jolla, CA), AZ Therapeutics (Cambridge, MA), Eisai (Tokyo, Japan), and consulting fees from Biogen (Cambridge, MA), Amylyx Pharmaceuticals, (Cambridge, MA), Mitsubishi Tanabe Pharma America (Jersey City, NJ), MediciNova (La Jolla, CA), and Tsumura (Tokyo, Japan). S.J.B. consults for Exicure (Chicago, IL), Clene (Salt Lake City, UT), KorroBio (Cambridge, MA), Faze Medicine (Cambridge, MA), and NeuroCures Foundation. T.M.M. received consulting fees from Ionis, Disarm Therapeutics, Cytokinetics and has licensing agreement with Ionis and C2N, and serves on the advisory boards of Biogen and UCB. R.B. is the founder of Iron Horse Diagnostics (Phoenix, AZ) and consults for Mitsubishi Tanabe Pharma America (Jersey City, NJ), Takeda (Tokyo, Japan), Aural Analytics (Scottsdale, AZ), and NeuroCures Foundation. J.K.W. is an ad hoc consultant for BridgeBio and Flagship Pioneering, and is on the Scientific Advisory Board of PepGen. L.P. is a consultant for Expansion therapeutics. R.H.B. consulted for Wave Life Sciences and is co-founder of Apic Bio. F.-B.G. received an honorarium from Alkermes company. The remaining authors declare no competing interests.
