## [Peer Review File · Nature Communications]

Reviewers' Comments:

Reviewer #1:

Remarks to the Author:

The manuscript by Krishnan et al describes two novel immunoassays for quantification of poly(GA) and poly(GR) levels in CSF. Previously poly(GP) has been quantified in CSF and e.g. shown to decrease in a C9ORF72-ALS patient after treatment with an ASO targeting the aberrant C9ORF72 transcript (Tran et al, Nat Med, in press). In the present paper, the authors show that also CSF levels of poly(GA) and poly(GR) decrease after ASO treatment in the same patient, and that this decrease might occur already within 6 weeks. The manuscript is interesting but there are several concerns:

1) The immunoassays could be better characterized following previously established guidelines, which e.g. include testing of dilution linearity (Andreasson et al, doi.org/10.3389/fneur.2015.00179). Further, intra-assay and inter-assay CVs should be given for non-spiked CSF samples obtained for appr. 20 mutation carriers.

2) Quite few patients are included in the statistical analyses studying the associations with clinical symptoms. For example, the differences between pre-symptomatic (n=10) and symptomatic patients (n=25) in Fig 2D and Fig3D could likely be significant if a more appropriate number of patients had been studied. Further, it is unclear why not the same patients are included in Fig 2 and Fig 3, and why certain cases were excluded. If several mutation carriers had undetectable CSF levels of poly(GA) and poly(GR) that should be clearly stated in the results section and figure legends.

3) The faster decrease in CSF levels of poly(GA) and poly(GR) versus poly(GP) in a single patient treated with an ASO is quite speculative and it needs to be confirmed in at least 5-10 patients.

Reviewer #2:

Remarks to the Author:

A very clearly written methods paper. The authors rightly emphasise that the incremental development here is an assay that shows target engagement only, based on our current understanding of C9orf72 ALS pathogenesis. They might make the specific point that this is still very important because decisions around therapeutic effects of ASO-type therapy might require prolonged studies (c.f. recent tofersen data showing early neurofilament drop but no change clinically at 6 months), and so it will be vital to show that any negative outcome is not attributable to lack of target engagement.

One point is that it is much more meaningful to show a relationship with clinical disease activity using *rate* of ALSFRS-R decline i.e. ALSFRS-R drop divided by disease duration. Absolute ALSFRS-R values are less consistent as a patient may have reached a given level of disability in years versus months in some cases for example. The standard formula for rate is: (48 minus ALSFRS-R at sampling / months from first motor weakness at sampling). NB to be useful it must be anchored to symptom onset and not date of diagnosis, the latter being too variable.

Reviewer #3:

Remarks to the Author:

The most common genetic cause of ALS and FTD is a repeat expansion of GGGGCC in the first intron of C9orf72. RNAs transcribed from that expanded repeat can be translated into dipeptide repeat proteins. Some of these dipeptide repeats have been found to be toxic. Some evidence has demonstrated that they can be detected in the CSF of people who harbor these mutations and are living with either or both of these diseases.

Here the authors develop assay systems for the detection of dipeptides polyGA and polyGR in the CSF from people with C9orf72-ALS/FTD. They then apply the assay system to samples from people

with the same.

The authors developed a Meso Scale Discovery (MSD) based assays to detect poly(GR) and poly(GA).

This work is very important and the results are noteworthy.

- 1) While an assay system had previously been developed to detect poly(GP) in CSF of people with C9orf72 repeat expansions, poly(GP) has not been demonstrated to be particularly toxic in any cell or animal experimental systems. GR and GA are likely to be more relevant to disease.
- 2) Development of sensitive assay systems that meet typical FDA criteria for confidence suggests that this system could be used in clinical development.
- 3) The fact that people who harbor mutations also harbored polyGA and polyGR in their CSF is an important finding considering that there were virtually no levels of either detected in healthy controls.
- 4) The fact that levels decreased following treatment with an ASO targeted at expanded C9orf72-RNA indicates that this could be a very meaningful measure of pharmacodynamic effects of that type of therapeutic and other therapies aimed at reducing production of dipeptide repeats.

The methodology is sound, but the paper would benefit if it addressed a few outstanding questions:

- 1) The assay systems rely on use of 8mer (polyGR) and 60mer (polyGA) in their standard curves. However, the size of the dipeptides in CSF remains unclear and may vary. The study would be stronger if it assessed the effects of different size dipeptides on spike/recovery percentage.
- 2) Samples were concentrated using 3k filter centrifugation. GR-8mer should only be approximately 2kDa. Is there risk that smaller dipeptides will flow through?
- 3) With regard to GR, there is some evidence that methylation state of the dipeptides may be important to their pathogenicity. The study would benefit if the authors assessed whether asymmetric or symmetric dimethylation of arginine residues impact its ability to be detected in the assay system.

While these points may not all be addressable within the scope of this study, they should be discussed.

This manuscript would benefit from minor revision.

Point-to-Point Responses

Reviewer 1

1. *The manuscript by Krishnan et al describes two novel immunoassays for quantification of poly(GA) and poly(GR) levels in CSF. Previously poly(GP) has been quantified in CSF and e.g. shown to decrease in a C9ORF72-ALS patient after treatment with an ASO targeting the aberrant C9ORF72 transcript (Tran et al, Nat Med, in press). In the present paper, the authors show that also CSF levels of poly(GA) and poly(GR) decrease after ASO treatment in the same patient, and that this decrease might occur already within 6 weeks. The manuscript is interesting.*

We thank the reviewer for the concise summary and positive comments on our manuscript.

2. *The immunoassays could be better characterized following previously established guidelines, which e.g. include testing of dilution linearity (Andreasson et al, doi.org/10.3389/fneur.2015.00179).*

We thank the reviewer for this constructive suggestion. For the poly(GR) immunoassay, we previously validated the assay range, the concentration-response relationship in sample matrix, and reproducibility by standard-curve analysis and spike-and-recovery experiments (Figure 1). As suggested by the reviewer, we did additional experiments to characterize the immunoassays, including the linearity and specificity of the dose-response, which are presented in revised Figure 1. Moreover, we cited Andreasson et al. (2015) in the revised manuscript.

It was unclear whether the reviewer also suggested performing these characterizations for the GA immunoassay as this assay utilizes the same antibody combinations used to detect poly(GA) levels in several prior publications, which we cited in the manuscript (Almeida et al., 2019; Nguyen et al. *Neuron*. 2020; Krishnan et al., 2020; Sonobe et al., 2021). Furthermore, the assay has undergone full qualification for evaluating CSF samples in clinical trial and is thus appropriately characterized. Biogen is currently using this assay to evaluate poly(GA) levels in patient samples from ongoing clinical trials. After obtaining legal approvals from Biogen, we have provided the GA immunoassay qualification data to the editor and reviewers only as a confidential document which will hopefully alleviate any concerns about the assay and the robustness of any results in our manuscript.

3. *Further, intra-assay and inter-assay CVs should be given for non-spiked CSF samples obtained for appr. 20 mutation carriers.*

Upon the reviewer's suggestion, we included the intra-assay CVs for all 25 mutation carriers. Unfortunately, we no longer have enough material to determine inter-assay CVs for these 25 mutation carriers. Therefore, we would have to request large quantity of additional precious CSF samples. However, we included the inter- and intra-assay CVs of three C9 samples originally used as a quality control in our runs. These results are now presented in Supplementary Figure 1.

4. *Quite few patients are included in the statistical analyses studying the associations with clinical symptoms. For example, the differences between pre-symptomatic (n=10) and symptomatic patients (n=25) in Fig 2D and Fig3D could likely be significant if a more appropriate number of patients had been studied. Further, it is unclear why not the same patients are included in Fig 2 and Fig 3, and why certain cases were excluded. If several*

mutation carriers had undetectable CSF levels of poly(GA) and poly(GR) that should be clearly stated in the results section and figure legends.

We agree that it would be helpful to test more samples to increase the power of our statistical analysis. However, these CSF samples are very precious and we tested all the samples we were able to obtain with our best efforts. In response to the reviewer's comment, we added the following sentence to the Results section: "Additional CSF samples, especially from pre-symptomatic patients, are needed to further confirm this result."

It is important to note that Poly(GA) and poly(GR) assays have different sensitivities, and the relative poly(GA) and poly(GR) levels in a given CSF sample do not always correlate. Thus, some CSF samples have detectable poly(GR) but not poly(GA), and vice versa. For these reasons, the samples assessed in Figure 2 and Figure 3 are overlapping but not identical and include 17 samples in which both GA and GR were detected. We also clearly stated in the Results section that "We obtained 41 CSF samples from symptomatic *C9ORF72* patients and 14 CSF samples from pre-symptomatic *C9ORF72* carriers. For 25 symptomatic patients that had poly(GR) levels within the linear range, poly(GR) levels in the CSF did not significantly correlate with ALSFRS-R scores (Supplementary Fig. 2a) or the rate of ALSFRS-R score decline (Fig. 2a)." "CSF samples from 10 pre-symptomatic *C9ORF72* patients had poly(GR) levels within the linear range and did not differ significantly with symptomatic patients (Fig. 2d). Additional CSF samples especially from pre-symptomatic patients are needed to further confirm this result." Similar information is presented in the figure 2 legend: "25 out of 41 symptomatic CSF samples and 10 out of 14 pre-symptomatic CSF samples had poly(GR) levels within the linear range of our standard curves and are presented here." Figure 3 legend: "The samples assessed in Figure 2 and Figure 3 are overlapping but not identical and include 17 samples for which both GA and GR were detected".

We like to note that none of the CSF samples obtained from the NEALS Consortium had a poly(GR) level below the linear range of our standard curve. This suggests that the way NEALS Consortium collects, processes, handles or stores the CSF samples protects sample integrity, at least for poly(GR). It will be important to determine whether this observation holds true as more CSF samples are assessed from the NEALS Consortium as well as other sources of CSF.

5. The faster decrease in CSF levels of poly(GA) and poly(GR) versus poly(GP) in a single patient treated with an ASO is quite speculative and it needs to be confirmed in at least 5-10 patients.

We agree that it would be good to confirm this result in additional *C9ORF72* patients treated with ASOs. However, as the reviewer will surely understand, it is currently impossible for us to obtain such samples—a limitation acknowledged in the original manuscript. For instance, we stated that this result needs to be confirmed in additional patients CSF samples (Results, p. 7; Discussion, p. 8). After considering the reviewer's comment, we changed a sentence in the Abstract from "CSF poly(GA) and poly(GR) levels decreased rapidly within 6 weeks" to "CSF poly(GA) and poly(GR) levels decreased approximately 50% within 6 weeks".

Reviewer 2

1. A very clearly written methods paper. The authors rightly emphasise that the incremental development here is an assay that shows target engagement only, based on our current understanding of C9orf72 ALS pathogenesis. They might make the specific point that this is still very important because decisions around therapeutic effects of ASO-type therapy might require prolonged studies (c.f. recent tofersen data showing early neurofilament drop but no change

clinically at 6 months), and so it will be vital to show that any negative outcome is not attributable to lack of target engagement.

We thank the reviewer for the positive comments. As suggested, we have highlighted the importance of demonstrating target engagement with our assays in prolonged studies. We now state in the Discussion that “these biomarkers could be used to assess correlations between improved clinical features and loss of DPR proteins, and in prolonged clinical trials, they could be used to show that when a lack of efficacy or a negative outcome occurs, it is not due to the lack of target engagement.”

*2. One point is that it is much more meaningful to show a relationship with clinical disease activity using *rate* of ALSFRS-R decline i.e. ALSFRS-R drop divided by disease duration. Absolute ALSFRS-R values are less consistent as a patient may have reached a given level of disability in years versus months in some cases for example. The standard formula for rate is: (48 minus ALSFRS-R at sampling / months from first motor weakness at sampling). NB to be useful it must be anchored to symptom onset and not date of diagnosis, the latter being too variable.*

We thank the reviewer for this constructive comment. Figure panels in which the rate of decline of ALSFRS-R scores is presented have been revised accordingly. The panels showing ALSFRS-R values have been moved to a supplementary figure.

Reviewer 3

1. This work is very important and the results are noteworthy. 1) While an assay system had previously been developed to detect poly(GP) in CSF of people with C9orf72 repeat expansions, poly(GP) has not been demonstrated to be particularly toxic in any cell or animal experimental systems. GR and GA are likely to be more relevant to disease. 2) Development of sensitive assay systems that meet typical FDA criteria for confidence suggests that this system could be used in clinical development. 3) The fact that people who harbor mutations also harbored polyGA and polyGR in their CSF is an important finding considering that there were virtually no levels of either detected in healthy controls. 4) The fact that levels decreased following treatment with an ASO targeted at expanded C9orf72-RNA indicates that this could be a very meaningful measure of pharmacodynamic effects of that type of therapeutic and other therapies aimed at reducing production of dipeptide repeats.

We thank the reviewer for the highly positive comments on our manuscript.

2. The assay systems rely on use of 8mer (polyGR) and 60mer (polyGA) in their standard curves. However, the size of the dipeptides in CSF remains unclear and may vary. The study would be stronger if it assessed the effects of different size dipeptides on spike/recovery percentage.

We agree that the size of the dipeptides in CSF is unknown. However, it is quite difficult to identify the entire range of DPR lengths in CSF since DPR proteins of different length may be synthesized in patient neurons and each DPR protein may be processed into smaller fragments. Thus, it would be extremely challenging, if not impossible, to test different size DPR proteins extensively. Our goal here is not to measure the *concentration* of each DPR protein of a certain size. By using (GR)₈ and (GA)₆₀ in our standard curves, we measure the amount of total poly(GR) and poly(GA) equivalent to the epitopes detected in (GR)₈ and (GA)₆₀ based standard

curves. As a result, we can quantitate, for example, the changes in poly(GR) and poly(GA) levels within each individual. Indeed, in the legend for Figure 2, we state that “Poly(GR) concentrations are presented as values equivalent to (GR)₈ peptide and not as an absolute poly(GR) concentration since the size of poly(GR) in CSF is unknown.”

In response to the reviewer’s comment, we revised the Methods section and made it clear that the sizes of DPR proteins in CSF are unknown and that we used standard curves to measure the amount of total poly(GR) and poly(GA) equivalent to the epitopes detected in (GR)₈ and (GA)₆₀.

3. Samples were concentrated using 3k filter centrifugation. GR-8mer should only be approximately 2kDa. Is there risk that smaller dipeptides will flow through?

We would like to clarify that 3-kDa filter centrifugation was used to concentrate patient CSF samples but not used to generate (GR)₈-based standard curves. Healthy people have up to 20–25 G₄C₂ repeats, whereas *C9ORF72* patients can have hundreds or thousands. Thus, we do not expect that pathological poly(GR) proteins will be smaller than 3 kDa.

Upon the reviewer’ comment, we revised our Methods section to point out that after 3-kDa filter centrifugation, very small dipeptides will not be captured, however we do expect pathological poly(GR) proteins to be captured as they are substantially larger than 3 kDa.

4. With regard to GR, there is some evidence that methylation state of the dipeptides may be important to their pathogenicity. The study would benefit if the authors assessed whether asymmetric or symmetric dimethylation of arginine residues impact it's ability to be detected in the assay system.

We thank the reviewer for raising this interesting question. Substantial efforts are needed to address this question experimentally and we are very interested in collaborating with experts on DPR protein modifications to systematically investigate the issue and then publish the results in a separate paper.

In response to the reviewer’s comment, we revised our Discussion section to point out that additional studies are needed to investigate post-translational modifications of DPR proteins (e.g., methylation) and the effects of such modification on the detection of DPR proteins in CSF samples. We now state that “additional studies are needed to investigate post-translational modifications of DPR proteins (e.g., methylation) and their effects on their neurotoxicity and detections in CSF samples.”

5. While these points may not all be addressable within the scope of this study, they should be discussed.

We thank the reviewer for his/her understanding. We have discussed or clarified all these points in the revised manuscript.

Reviewers' Comments:

Reviewer #1:

Remarks to the Author:

If I understand the newly added text in the results correctly, poly(GR) could be reliably detected in 25 out of 41 (61%) of symptomatic cases and 10 out of 14 (71%) of presymptomatic cases.

Similar numbers were observed for poly(GA). Would not this be a great limitation if to be used to determine target engagement, i.e., that the CSF levels in many participants cannot reliably detected even before treatment? This should be discussed in the paper.

Further, the authors note that "none of the CSF samples obtained from the NEALS Consortium had a poly(GR) level below the linear range of our standard curve", but this is not the case for many of the other cohorts. Therefore, there must be unknown pre-analytical factors substantially affecting the CSF levels either when collecting, processing, or storing the samples. Again, this should be discussed in the paper.

Reviewer #2:

Remarks to the Author:

Comments addressed

Reviewer #3:

Remarks to the Author:

Through this revision, the authors have addressed my comments and concerns.

Reviewer 1

If I understand the newly added text in the results correctly, poly(GR) could be reliably detected in 25 out of 41 (61%) of symptomatic cases and 10 out of 14 (71%) of presymptomatic cases. Similar numbers were observed for poly(GA). Would not this be a great limitation if to be used to determine target engagement, i.e., that the CSF levels in many participants cannot reliably detected even before treatment? This should be discussed in the paper. Further, the authors note that "none of the CSF samples obtained from the NEALS Consortium had a poly(GR) level below the linear range of our standard curve", but this is not the case for many of the other cohorts. Therefore, there must be unknown pre-analytical factors substantially affecting the CSF levels either when collecting, processing, or storing the samples. Again, this should be discussed in the paper.

We thank the reviewer for this thoughtful discussion. In the revised Discussion section, we now state the following: “Our study has some limitations that are worth discussing. First, several CSF samples from the Biogen cohort exhibited poly(GR) or poly(GA) signals outside the linear range of our standard curves and therefore could not be included in our analyses described here. In contrast, we were able to measure poly(GR) levels for all of the NEALS cohort CSF samples. Given this observation, we speculate that there are likely to be unknown factors that contribute to CSF DPR protein levels and may therefore account for these differences. For instance, it is possible that the collection, processing and storage of the CSF samples is a key aspect to sample integrity and that different clinical sites have slightly modified protocols that may affect sample quality (e.g. the length of collection time, the person collecting samples, collection vessels, time and site of puncture, etc.). Consistent with this notion, others have shown that variations in sample handling can affect the measurable levels of prion protein in the CSF³⁹. Nevertheless, given that detectable levels of both poly(GA) and poly(GR) were observed in a majority of cases, in addition to further improvement of assay sensitivity, one way to overcome this limitation in a clinical trial setting is to first ensure the quality of CSF samples and then pre-screen trial participants based on detectable CSF DPR protein load prior to enrollment. With such patient selection criteria used in trial designs, appropriate assessment of candidate therapeutics will become more robust.” The second limitation is same as the one we presented previously regarding sample sizes.